# Experience with a Self-Management Education Program for Adolescents with Type 1 Diabetes: A Qualitative Study

**DOI:** 10.3390/nursrep15010022

**Published:** 2025-01-14

**Authors:** Marília Costa Flora, Luísa Barros, Maria Isabel Dias da Costa Malheiro

**Affiliations:** 1Nursing Research, Innovation and Development Centre of Lisbon (CIDNUR), University of Lisbon, Nursing School of Lisbon, 1600-190 Lisbon, Portugal; mmalheiro@esel.pt; 2Department of Child and Adolescent Nursing, Nursing School of Coimbra (ESEnfC), 3004-011 Coimbra, Portugal; 3Faculty of Psychology & Research Center for Psychological Science (CICPSI), Lisbon University, 1649-013 Lisbon, Portugal; lbarros@psicologia.ulisboa.pt; 4Department of Child and Youth Nursing, Nursing School of Lisbon, 1600-190 Lisbon, Portugal

**Keywords:** adolescent, diabetes mellitus, type 1, focus groups, self-management, program evaluation, nursing research

## Abstract

**Background/Objectives**: Adolescents with type 1 diabetes face complex challenges associated with the disease, underscoring the importance of developing self-management skills. This study examined participants’ perspectives on a type 1 diabetes self-management education program. **Methods**: Focus group interviews were conducted with 32 adolescents with type 1 diabetes who participated in the program and six expert patients. Both thematic analysis and content analysis were conducted using NVIVO software, version 1.6.1. **Results**: Two dimensions emerged: expert patient roles and program evaluation. Expert patient roles were viewed positively, with an emphasis on responsibility, sharing experiences, and being a role model. Program evaluation emphasized peer-to-peer sharing and educational sessions, with increased knowledge of the disease and management strategies. Expert patients also benefited from the program by developing a sense of responsibility, serving as role models for adolescents, and improving their disease self-management. **Conclusions**: The adolescents emphasized that the program was a learning tool and the expert patient’s view of their role emerges, highlighting modelling as a facilitator of learning and of the responsibility and commitment of the mentors. This study reinforces the benefits of peer-to-peer interaction in a camp setting, including rich experiences.

## 1. Introduction

Self-management education programs are essential for empowering people with chronic diseases [1] and are considered key elements in the management of type 1 diabetes (T1D) [2,3]. In addition to symptom management and treatment, they promote psychosocial and behavioral aspects that improve the well-being and quality of life of people with T1D [4,5], namely through blood glucose control [6].

Adolescents experience a process of progressive transition towards autonomy as they develop and build autonomy. However, when the first episode of T1D occurs in adolescence, the adolescent is also experiencing a health/illness transition. At this stage of the study, we are considering the hypothesis that a summer camp with a self-management education program will help adolescents with T1D become autonomous in managing their disease, in accordance with Afaf Meleis’ Transitions Theory [7].

Lorig and Holman were pioneers in the development and implementation of self-management programs for adults with chronic diseases using expert patients, including the Expert Patients Program (EPP) [8], which resulted in fewer hospital admissions and complications and improved therapeutic adherence, quality of life, and well-being [9]. Malheiro developed a self-management education program for adolescents with spina bifida (SEPASB) based on Lorig and collaborators’ EPP [10]. This program had a significant impact on promoting self-management skills, suggesting that it could be adapted for adolescents with other chronic diseases [10].

The self-management education program for adolescents with T1D—Lay Led Diabetes Education for Adolescents with Type 1 Diabetes (LayLeDU DM1) was developed and adapted from the SEPASB. This program uses expert patients, that is, young people with T1D who are experts in the self-management of their disease and, after training, take on the role of educators and act as facilitators in developing the self-management skills of their peers [10]. The expert patients underwent training before the camp and actively participated in the creation of case scenarios for role-playing exercises. Expert patients facilitate adolescents’ autonomy by implementing psychoeducational strategies, including questioning, problem-solving, modeling, and the supervision of procedure checklists and action plans (completed at the end of each session). They also serve as a liaison between adolescents and health professionals. The program uses a tutoring system, where older adolescents are tutors to younger ones and are supervised by expert patients. These tutors are selected based on age, rather than self-management skills [10].

The program includes daily sessions that follow a predetermined protocol: Brief introduction to the topic related to each session; Questioning to identify the participants’ knowledge about the topic; Brainstorming to identify the problems mentioned by the adolescents related to the topic; Problem-solving to discuss possible solutions to the most common problems; Role-playing (simulation of two or three problems previously identified by the expert patients); Short lesson (summarized presentation of key aspects and videos related to the topic); and Action plan, in which the participants prepare a plan and commit to change at least one behavior related to the topic of the session [10].

An individual logbook with a checklist of daily procedures was developed to monitor daily procedures and record individual goals after each session and was given to each adolescent at the first session of the program. According to Lorig, writing in a personal diary allows for awareness, action, and process monitoring [11]. A checklist of three procedures (ketone correction, meal timing, and interstitial glucose monitoring) is provided in the logbook, and adolescents mark each step they take in performing these procedures on the detailed checklist each day. This activity helps adolescents perform the procedure properly, and daily repetition helps them memorize it so that it becomes routine [10]. At the end of each session, each participant can take their “individual logbook” and fill in the goals they have set, with their tutor’s validation.

Sessions were organized as follows: (1) The program structure and methods were explained, tutor/student pairs were distributed, and individual logbooks with the action plan and goal setting were provided; (2) Issues related to the etiology and causes of T1D were addressed, and participants were informed about the complications of poorly controlled diabetes; (3) Information was provided on how to make food choices on special days, and concepts of diet/exercise and insulin therapy were discussed; (4) Insulin was administered by continuous subcutaneous insulin infusion (CSII); (5) A motivational exercise video was shown to raise awareness and motivate adolescents with T1D to exercise and positively reinforce adolescents who exercise regularly with high-performance levels.

Topics were selected based on the results of an exploratory study designed to identify factors that facilitate or inhibit the development of autonomy in adolescents from the perspective of young adult experts with T1D and their parents [12], as well as the synthesis of evidence on interventions that promote self-management skills in adolescents with T1D.

In Portugal, the reviewed literature highlights the initiatives of the Portuguese Diabetes Association, which organized holiday camps for adolescents with T1D in recent years, with data from these camps disseminated in various reports [13,14]. However, there remains a significant gap in the available information regarding the impact of these camps on self-management.

The Lay LeDU DM1 program was implemented in a summer camp over six days. A qualitative evaluation of this experience was carried out at the end of the program using three focus group interviews with the adolescents (male, female, and expert patients).

This study aimed to describe and analyze the participants’ and expert patients’ perspectives on the program.

## 2. Methods

A qualitative study was conducted using focus group interviews. Considering that adolescents experience a developmental transition characterized by profound physical, cognitive, and social changes, the conceptual framework used was Afaf Meleis’ Transitions Theory [15]. They also experience a health and illness transition toward greater autonomy in managing the illness. The educational intervention strategies employed in the program were grounded in Bandura’s sociocognitive theory and emphasized key concepts such as self-efficacy, self-regulation, and observational/modeling learning [16]. The manuscript was written following the COREQ—Consolidated Criteria for Reporting Qualitative Research [17].

A purposive sample comprised adolescents recruited through a diabetes association covering the Lisbon metropolitan area, Southern Portugal, and the Azores and Madeira archipelagos. Adolescents with T1D were selected to participate in the summer camp based on the following criteria pre-established by the health team: (1) adolescents with T1D; (2) with glycated hemoglobin (HbA1c) levels higher than 8% at the last health examination; and (3) attending diabetes consultations in the association.

Parents and adolescents were contacted by a team’s nurse by telephone, after which information and informed consent forms were sent by email and signed on the first day of the camp. All adolescents who attended the camp were invited and agreed to participate in the focus group interviews. Authorization was obtained from the Ethics Committee of Diabetic Association. Parents and participants signed informed consent N.114/2017. The study was conducted in August 2019.

The team that accompanied these adolescents was composed of two doctors (an endocrinologist and a pediatrician), two nurses who specialized in DM1, one nutritionist, one psychologist, and one operational assistant. The team also included four adolescents who, in addition to lay LEDs, also took on the role of monitors who participated with the role of monitors who participated as group leaders and liaisons between the group and the health team and two Lay Leds (young adults who are experts in the management of T1D and members of the Youth Center of Diabetes Association) who participated in the planning and development of the activities.

Focus group interviews were conducted with three groups of participants: Group 1—male adolescents (*n* = 16), Group 2—female adolescents (*n* = 16), and Group 3—expert patients (*n* = 6, three male and three female young adults). Male and female participants were in separate groups. This strategy, derived from the original study, showed favorable results as it allowed adolescents of the same sex greater freedom to share personal experiences. The interviews were videotaped with the consent of the participants. They took place on the last day of the summer camp and lasted approximately one hour each. The rooms in which the interviews were conducted provided privacy and comfort, and participants were seated in chairs arranged in a circle to facilitate communication.

The interview script for the adolescents and expert patients was prepared according to the objectives of the study [18] and included the following questions: “Is this your first time at the camp? How did you feel (icebreaker)? What do you think of the summer camp? What do you think of the program? What do you think of the sessions (topics, duration, moderators, and methods)? What do you think of the tutoring system?” The interview ended with an open-ended question: “Is there anything else you want to add?” (to ensure that there was nothing more to add to what had already been said).

The principal investigator, a pediatric nurse specialist in child health, conducted the interviews after taking part in all camp activities and having developed a relationship of trust with the adolescents. A nurse from the multi-professional team and a diabetes expert participated as an assistant moderator. He was a nurse, and usually attended to them at the diabetes consultations.

Throughout the interview, the moderator made notes that she considered relevant. The interviews were transcribed in full, and significant nonverbal communication was recorded. The focus group discussions were first transcribed by the first author (MCF) confirmed for correctness by the third author (MIDCM), and checked by the study supervisors (MIDCM, LB). Researchers’ triangulation was applied during the entire process.

Data were analyzed based on Bardin’s content analysis technique [19], with some particularities related to the focus group data analysis technique, which seeks to give more consistency to the results by Extensiveness, i.e., the number of different participants who talked about a given topic or agreed through nonverbal communication, and Frequency, which refers to the number of times a given topic comes up in the discussion [20].

For the content analysis, the three steps proposed by Bardin were followed: pre-analysis, which resulted in categorization and coding; material exploration, during which the categories formed in the previous stage were analyzed; and finally, result processing, including inference and interpretation. NVIVO software version 1.6.1 was used for data analysis and categorization.

## 3. Results

The expert patients were between 15 and 21 years old, consisting of three boys and three girls. All of them had previously attended a summer camp, and, for insulin administration, only one of them used MDI (multiple dose injection therapy) while the others used CSII. Table 1 shows the sociodemographic and clinical characteristics of the adolescents who participated in the program.

An initial characterization was developed considering the 100 most cited words in the focus group interview related to the program implemented in the summer camp. The words most frequently used by the participants were highlighted to illustrate the meaning they attributed to the program. The results showed that the participants mainly referred to the program as an interesting experience involving responsibility (Figure 1).

Data analysis regarding the Lay LeDU DM1 program revealed two distinct dimensions among the group of expert patients related to their roles and program evaluation, whereas only the latter was identified among the group of participants.

### 3.1. Evaluation of the T1D Self-Management Education Program (Expert Patients and Participants)

Two Categories Emerged Related to the Evaluation of the Program from the Perspective of the Participants and the Expert Patients: Positive and Negative Aspects.

*Program evaluation from the perspective of expert patients and participants: positive aspects*.Figure 2 shows some positive aspects of the program from both groups, namely the increase in knowledge about T1D management: “I think I know more now (for) a person who started wearing the pump this year (…) I didn’t know anything (…) and now I know and I think I’m going to use that knowledge and it’s going to help me get better … I learned that here.” (F9) “We learn how to do things well here.” (F13) Another positive aspect was the topics addressed in the sessions: “The sessions are interesting because they address topics that maybe aren’t normally discussed in consultations or a daily basis, and some things are always discussed in more depth because they’re all together, they discuss the things on the agenda and they exchange ideas, so it’s good for them.” (EP1) “I liked the topics of the sessions (…).” (EP4) “The sessions are important, they’re very useful, I found them interesting and important.” (F9).
Figure 2Program evaluation from the perspective of expert patients and participants: positive aspects. * Number of participants who report the topic; ** Number of times that the topic appears in the interview corpus.
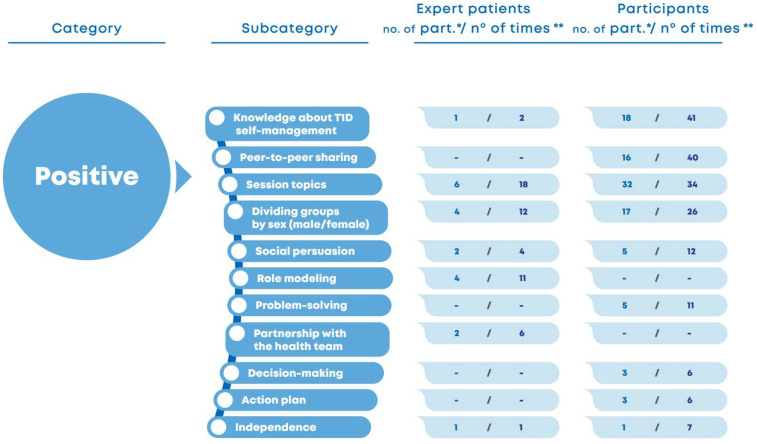

Another positive aspect relates to peer-to-peer sharing and the social interactions that come with it: “I liked interacting with everyone the most.” (M9) “It was getting to know new people and their story with diabetes.” (F11) “I think sharing can be important, to understand that we’re not alone …” (M3).Participants also talked about the fact that the groups were divided by gender: “At first, when we were divided into boys and girls, I was afraid (…) But I realized, like, it’s easier for the boys, I speak for the boys, girls I don’t know, but at least it was easier for us to talk about things only with boys.” (M16) “The boys are different, they feel free to talk, they talk about everything without any worries, some more than others, I’m not going to say that in the boys’ group there aren’t some who (are) more reserved, but they’re great at talking and when the atmosphere gets serious they also get serious, there’s always that moment of fun, but the topic is always serious, I like that.” (EP5) “It was kind of interesting that it was boys and girls, I think it was good that it was just boys and girls because with the boys we talk about everything, and with the girls, we’re not so comfortable talking about those things (…).” (M8).The expert patients highlighted the tutoring system: “I also liked the tutoring system (as the implementation strategy) (…). I think they can feel encouraged to maybe become expert patients one day and I think that they should work towards it! I think the project itself is good, I liked it.” (EP2) “(…) It makes us proud of our students. We are helping them and they are helping other people at the same time.” (EP4) “I’m proud to listen to them because we were also participants and now that we are (expert patients), we see things differently and we can help them.” (EP5) Both expert patients and participants referred to processes of social persuasion: “(…) there are other people going through the same thing, and if some succeed, others will too.” (F5) “We can even use the strategy that those people use in that situation (…).” (EP2).With regard to self-management skills, problem-solving related to T1D management stood out: “I think I’ve learned to control it better at night, even if I’m not the one who wakes up, because I’ve woken up a lot.” (F1) “I think I’ve started to be more rigorous, like at home, for example, I usually take rough measures and here you have to be really rigorous…” (F13) “I think I’ve learned to manage insulin reductions better, to exercise better, and to correct hypoglycemia better.” (F5) They also considered the relationship established with the health team to be relevant: “We also have activities, the health team is also excellent, super fun, they jump in the water like us and everything!” (EP2) “It’s different, the consultations are scary because we’re scared when we’re not controlled, we go there scared and sure that we’re going to be scolded (…) but there’s very little time for that connection between people.” (EP2) Finally, the creation of an action plan was highlighted: “I’m going to start waiting before eating, which I had trouble doing at home, but I realized it wasn’t that hard.” (F3).Independence from parental supervision was another advantage of the camp: “In terms of diabetes, the camp contributes a lot to their growth, it gives them more freedom and teaches them a lot of things that sometimes they already know, but have never had the time to practice or because they are too dependent on their parents, the health team, or have never had experience with extreme sports.” (EP3).*Program evaluation from the perspective of expert patients and participants: negative aspects*.In the negative aspects category (Figure 3), the participants talked about the role of tutors: “In my opinion, it was quite a difficult role, I had to deal with difficult kids, but that’s the way it is, I had to learn how to reconcile diabetes with taking care of them.” “(…) I speak for myself, it was hard to meet, you know, my student (…) I think it’s difficult.” (F9) “The tutor/student system was very confusing in the beginning and also there wasn’t much time to do it …” (F10).
Figure 3Program evaluation from the perspective of expert patients and participants: negative aspects. * Number of participants who report the topic; ** Number of times that the topic appears in the interview corpus.
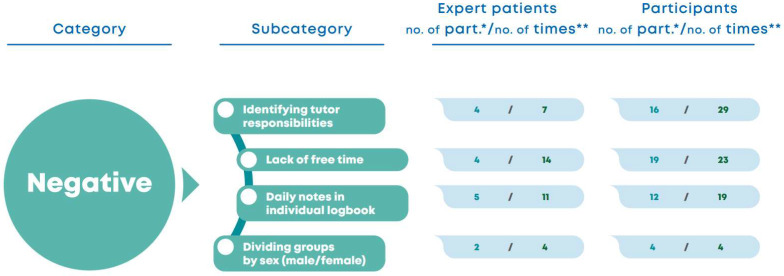

Another negative aspect was the high number of activities and the lack of free time for other activities: “Time is often limited and sometimes we run out of time to do other activities. I’m talking about time for everything.” (EP3) “For example, one time (…) we had some free time, we could choose between going swimming or playing basketball or soccer, I could use having more free time like that.” (M7).The daily record-keeping in the individual logbook was pointed out as a negative aspect: “This thing of filling in papers, we already get tired and then we also get up early … it is a bit boring.” (F2) “The paper seemed quite simple and to the point. They didn’t fill it in because they didn’t have time and were tired, at night they just want to go to bed.” (EP3) “I think we should keep the logging, as long as there’s always a specific time to do it.” (M7).Finally, the division of the groups by sex (male/female) was the source of controversy among some of the expert patients: “I don’t like being separated. Because I like us all being together (…) I like different things because, like I said, I really like having guy friends and interacting with them because, I don’t know, they’re different! (…) and sometimes it’s interesting to interact with them about those things.” (EP2) “(…) the fact that we’re limited to just 1 h per group instead of 2 h, which could be very dynamic (…) is one of the negative aspects of this division.” (EP3).

### 3.2. Expert Patients’ Roles

Two Categories Emerged Related to the Expert Patients’ Assessment of Their Roles in the Program: Positive and Negative Aspects.

*Expert patients’ assessment of their roles: positive aspects*.Figure 4 shows that the expert patients highlighted the positive aspects associated with their roles, namely responsibility: “(…) instead of taking care of one team, I take care of four. That is a huge responsibility.” (EP3) “I feel good, it’s a bigger responsibility, no doubt about it.” (EP2) They also indicated that being selected as expert patients was a recognition of their skills and qualifications by the health team: “I was very happy to be invited (…) What? Oh my God! I was screaming all over the house. I was really happy when I got the news! It’s an honor.” (EP2) “It was one of the biggest news I’ve had this year. I was very happy with the news, and I’m loving it (…).” (EP3).
Figure 4Expert patients’ assessment of their roles: positive aspects. * Number of participants who report the topic; ** Number of times that the topic appears in the interview corpus.
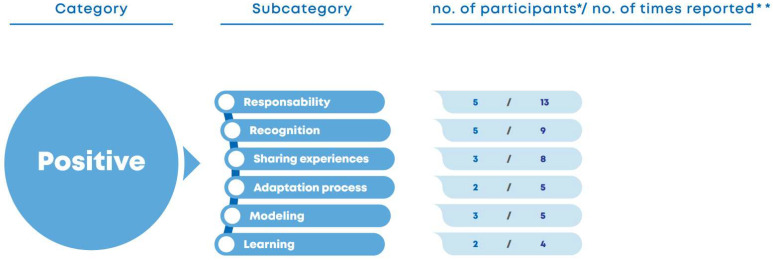

The sharing of experiences among the expert patients was also highlighted as a positive aspect: “It’s different sometimes to have people who understand what I’m doing, so this bond between us I think is why I think this week is always very intense because it’s something that brings us very close and because we share the deep things that we’ve experienced with the disease and everything. (…) I think having this disease helps us to release a little bit of what we’ve kept inside that we sometimes can’t talk about with our friends because they don’t understand us completely!” (EP2) “Sharing is very important.” (EP3).They referred to this experience as a moment of togetherness, sharing, and learning, which they believe facilitates the process of adapting to T1D: “We learned how we can improve our lives and adapt better to our disease. From now on, everyone will live better with the disease.” (EP6) “Accepting the disease is better than fighting it. (EP2) “(…) and to accept ourselves in spite of this disease because we are the same, it’s not a disease that defines us.” (EP2) They also recognized their importance as role models in the learning process of the younger ones: “They always tell us that the younger ones look up to us as examples, I could see that.” (EP5) “Being with other people helps because they see other people’s experiences, what they do and how they do it, they put it all together and do it better!” (EP1), as well as in their own learning: “But it’s good for the participants, for us to know more about diabetes because we’re all together and that helps a lot.” (EP1) (…) For them to understand that it must be done this way or that way, it has to be that way for them to understand. And we learn from that too.” (EP4).
*Expert patients’ assessment of their roles: negative aspects*
As for the negative aspects (Figure 5), they recognize that the intense sports activities and the responsibility of expert patients to measure the adolescents’ blood glucose levels during the night, together with the health team, were a physical overload: “The hardest part for me as a supervisor is having to get up at three and five in the morning, that’s the hardest part! Because I’m a heavy sleeper, so it’s always a little bit harder for me.” (EP2) “The hardest part is the responsibility because (…) there are a lot of kids, some really don’t know what they have to do, others know but don’t want to.” (EP1) They also refer the negative impact on their T1D management “Our levels have been a little bit more out of control this week because we’re more concerned about them, and well we take a backseat and only take care of ourselves when they’re done, and then everything gets out of control.” (EP1) “We forgot about ourselves a bit, but we had to!” (EP4) “We forgot about ourselves until the very end so that we could take care of them, that was our job.” (EP6).

## 4. Discussion

The analysis of the focus group interviews conducted with the participants and the expert patients revealed that, overall, the participants found the self-management education program to be an interesting experience, with a particular emphasis on peer-to-peer sharing of experiences and improving their knowledge about T1D management.

The expert patients were positive about the importance of their roles and the program’s contributions to their personal development. They emphasized the responsibility they felt in taking on these roles, albeit supervised by the health team. The expert patients reported that they felt recognized by the health team for giving them the responsibility of caring for a group of adolescents. They were happy to have the opportunity to help and teach younger adolescents. They saw it as a benefit because they had to demonstrate proficiency in managing their condition to effectively educate them. This proficiency required expert patients to be actively involved and take on the role of experts in managing their condition [10,21].

The importance of peer-to-peer experience sharing was also highlighted by both expert patients and participants. These findings suggest that the experience of interacting with peers who have the same condition increases opportunities for social interaction and allows for the sharing of everyday life experiences. The group is a pivotal relational space during adolescence, offering multiple opportunities for the cultivation of experiences and knowledge that contribute to the formation of an individual identity [22]. Support among peers with the same condition helps alleviate the suffering associated with diabetes and promotes the development of self-management skills [23]. This underscores the importance of peer-to-peer sharing in identifying common problems and exploring different strategies to address them.

Having diabetes means being different from your healthy peers and managing T1D requires complex care for which adolescents are not always prepared. A scoping review of 12 studies concluded that most adolescents feel different from their peers, which leads to social withdrawal [24]. A study examining the perceptions of mothers of young people with T1D about the benefits of camps pointed out that camps are a critical factor in the process of accepting the disease, as they provide an opportunity to make new friends with peers who have the same disease and allow adolescents to develop a more positive attitude about their disease and life in general [25].

In this study, adolescents highlighted the knowledge about T1D acquired at the camp. Therapeutic education involves transferring knowledge and the development of skills that promote behavior change [23,26]. Thus, therapeutic education is described as a process comprising organized activities of awareness, information, training, and psychological and social support that are designed to help the person with diabetes and their family understand the disease and treatments, so that they can participate in care, take responsibility for their health, and increase their autonomy [27,28].

The adolescents appreciated the sessions that complemented the other activities of the camp. In this way, promoting relational, helping, and sharing skills was combined with the systematization of knowledge. Several authors have recognized the advantages of complementing participation in educational sessions with recreational and sports activities, with psychological and clinical benefits and gains in the acquisition of self-management skills [26,29]. A 12-month study of an educational program for children and adolescents aged six to sixteen years that included various activities and workshops, culminating in a summer camp, showed a significant improvement in self-management knowledge and behaviors, demonstrating that the educational program increased participants’ knowledge [26].

With regard to the division of the group by sex (male/female), this strategy was well received by the majority of the participants, who felt more comfortable sharing their experiences with people of the same gender. Most participants recognized that they were less inhibited and could talk more openly about some aspects of their more private experiences. However, some participants felt this strategy was a limitation to a more inclusive peer-to-peer sharing of experiences. This strategy has already been implemented in the program for young people with spina bifida, with effective results in developing self-management skills [10].

The development of self-management skills was mentioned more often by participants than by expert patients. Participants were more likely to emphasize problem-solving, decision-making, and action planning. At the same time, expert patients, due to their level of expertise, will have developed these skills before, so they prioritize other aspects, like self-efficacy, which was mentioned in their responses. In a study that surveyed 4,562 caregivers of children and adolescents who attended a summer camp and collected data one week and one month after the camp ended, results showed that problem-solving skills increased by 17% at the end of the first week. One month later, participants were able to independently perform at least one self-management activity (monitoring diabetes, preparing insulin, administering insulin, and rotating the infusion site) [29]. Other studies have reported that implementing a program outside the context of a summer camp supports the development of self-management skills [23,26]. Similarly, in the study conducted by Malheiro, adolescents with spina bifida highlighted gains in self-management skills, with particular emphasis on the problem-solving strategy worked on in the program’s sessions, where the participants identified the problem and worked together to find solutions [10].

The adolescents and expert patients highlighted that the camp helped them become independent from their parents. A scoping review of 12 studies on the experiences of adolescents with T1D found that the development of autonomy depends on socioeconomic status, parents’ willingness to relinquish control, and adolescents’ ability to effectively manage T1D, among other factors [24]. The fact that these adolescents attend these camps without parental supervision and with the support of a multiprofessional team behind them provides an opportunity to develop autonomy. In a study evaluating continuous blood glucose monitoring systems during a summer camp for adolescents with T1D, the authors highlighted the positive impact of the camp on autonomy in managing T1D and adjusting to the disease [30].

Another negative aspect mentioned by some participants was the daily record-keeping of the procedures in the individual logbook at the end of the day under the tutor’s supervision to systematize them and facilitate their integration into daily routines since they had to complete the checklist every evening after a day of intense activity. According to Lorig, daily record-keeping helps to memorize and transform this activity into a routine [8]. Therefore, this activity should be reconsidered, and the schedule should be reformulated or modified in future programs [8].

The tutoring system was perceived as a negative aspect by the participants. They associated the system with their difficulty in understanding the role of the tutor and the fact that tutors and trainees were located in different dormitories, which made communication between them difficult, the participants suggest that, in future camps, tutors and trainees could be in the same dormitories to facilitate the process. Conversely, the expert patients perceived the tutoring system as beneficial and recognized it as a first step towards achieving the status of expert patients. In Malheiro’s study, the tutoring system was highly valued by the adolescents, particularly the tutors, who associated it with the responsibility of imparting knowledge to the younger generation. It was also perceived as a means of social advancement among peers [10].

During the research, some difficulties arose, such as the fact that the researcher was an outsider to the team, which meant she had to integrate and gain the team’s trust. As far as the team was concerned, there was some resistance to implementing the self-management education program, not least because they had an external team member participating in the fieldwork. We also consider the difficulty of fully implementing the SEPASB, the constraints encountered were mainly due to the impossibility of training the expert patients. Another possibly detrimental aspect was the reduction in the number of sessions, considering that the program was part of the usual summer camp planning, which led to the need to adjust to possible schedules and availability.

## 5. Conclusions

The focus group interviews’ analysis provided insight into participants’ and expert patients’ perceptions of the program, identifying both positive and negative aspects. Overall, the methods and psychosocial strategies were well accepted by the participants, who valued the tutoring system and the daily educational sessions.

The expert patients’ view of their roles demonstrates that role modeling facilitates learning, a sense of responsibility, and, above all, the recognition and appreciation of the health team for their ability to take on these roles. In turn, the participants valued experience sharing, social interactions, and the program’s focus on developing self-management skills.

The results of this study are relevant to the design of new programs in this area. This study enabled the development of knowledge regarding the design and operationalization processes of Lay LeDU DM1. The focus groups provided insights into the meaning attributed to the program by the adolescents and Lay Leds, with participants generally considering it an engaging experience.

We suggest expanding the program’s implementation to other contexts, such as schools or community groups, in addition to camps. The educational sessions and the topics covered were well-received by the adolescents and appear to have positively influenced their understanding of T1D.

Although some improvements and adjustments are needed, the adaptation of SEPASB seems to have been successful.

## Figures and Tables

**Figure 1 nursrep-15-00022-f001:**
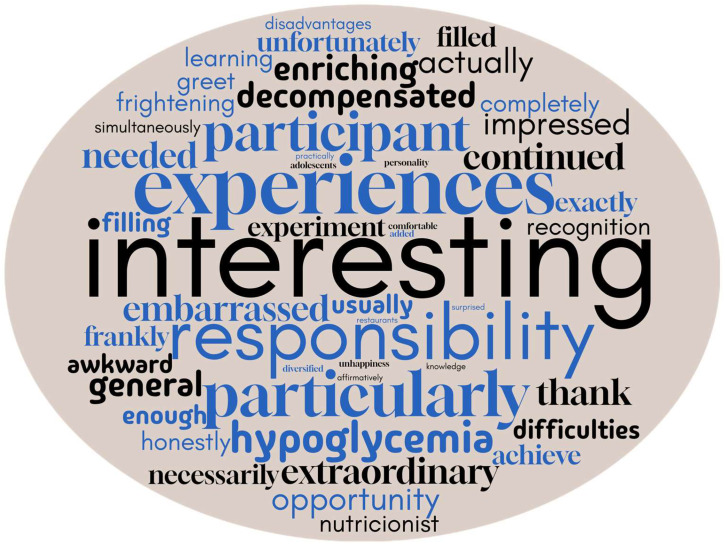
Word Cloud of the most cited words cited in the focus group interview related to the program implemented in a summer camp. Only the interviewees’ responses were used for the word cloud (NVIVO software version 1.6.1).

**Figure 5 nursrep-15-00022-f005:**
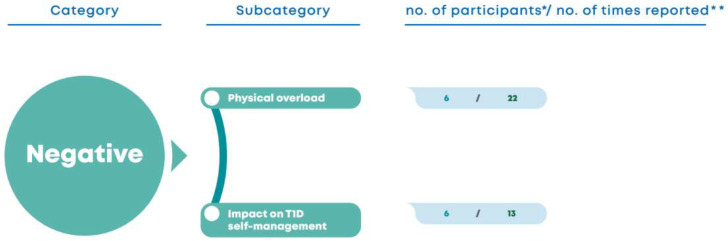
Expert patients’ assessment of their roles: negative aspects. * Number of participants who report the topic; ** Number of times that the topic appears in the interview corpus.

**Table 1 nursrep-15-00022-t001:** Adolescents’ sociodemographic and clinical characteristics (*n* = 32).

	$\bar{x}$ (*SD*)	Min–Max
Age	14.94 (1.39)	13–18
Age at diagnosis	8.11 (3.14)	1–14
HbA1c	8.17 (1.14)	6.30–11.40
	No.	**%**
Gender		
Male	19	59.4
Female	13	40.6
Age group		
13–15	22	68.8
>15	10	46.7
Time to T1D diagnosis		
<8 years	22	68.8
≥8 years	10	31.3
Insulin administration route		
CSII	23	71.9
MDI	9	28.1
First time participating in the camp		
No	13	40.6
Yes	19	59.4

CSII: Continuous Subcutaneous Insulin Infusion; MDI: Multiple Dose Injection Therapy.

## Data Availability

Data are contained within the article.

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
