# Peer review of "Experience with a Self-Management Education Program for Adolescents with Type 1 Diabetes: A Qualitative Study"

_nursrep, 2025, doi:10.3390/nursrep15010022_

Round 1

Reviewer 1 Report

Comments and Suggestions for Authors

Congratulations on the work you did. It was certainly interesting for the participants. As an added value, it would be important to publicise and promote the methodology among other teams. Type 1 diabetes is a silent disease that causes serious damage, especially to the microcirculation, with repercussions on future quality of life. Therefore, all strategies are welcome to help young people deal with the disease effectively.

Author Response

The authors thank the editor and the reviewers for the suggestions and analyses that were accepted, hoping they can contribute to improving the quality of the manuscript.

Comment 1: Title: 13 words. Uses words that are descriptors in health but does not mention the type of study

Response 1: We agree, changes were made and highlighted in yellow in the text. 

Comment 2: Consider: Experience with a self-management education program for adolescents with type 1 diabetes: a qualitative study

Response 2: We agree, changes were made and highlighted in yellow in the text. 

Comment 3: Abstract: KeyWords: Does not refer to any descriptor related to nursing Consider: Nursing research

Response 3: We agree, changes were made and highlighted in yellow in the text. 

Comment 4: Introduction - It presents a brief history of self-management projects for chronic pathology and the various changes made for different populations. It does not mention what is not known or the purpose of this study.

Response 4: We agree, changes were made and highlighted in yellow in the text. "

In Portugal, the reviewed literature highlights the initiatives of the Portuguese Diabetes Association, which organized holiday camps for adolescents with T1D in recent years, with data from these camps disseminated in various reports [12,13]. However, there remains a significant gap in the available information regarding the impact of these camps on self-management."

Comment 5: Methods and Materials - Was the program Lay LeDU DM1 developed as described in the introduction? The date of data collection is not displayed.

Response 5: The program was described in the introduction. The study was conducted in August 2019.

Comment 6: 

Some questions:

  • Was a literal translation of the contents into Portuguese?
  • What preparation is provided to expert patients?
  • How were the contents of the different sessions prepared?
  • Have normal camp activities been modified?
  • What is the role of health professionals?

Response 6: 

A literature review was conducted to identify interventions that promote self-management skills in adolescents with Type 1 Diabetes (T1D). To determine the themes of the sessions, a preliminary study was carried out using two focus groups aimed at understanding the perspectives of adolescents with T1D and their parents regarding the factors that facilitate or hinder the development of autonomy during adolescence.

Normal camp activities were no modified. 

The team that accompanied these adolescents was composed of two doctors (endocrinologist and pediatrician), two nurses who specialized in DM1, one nutritionist, one psychologist, and one operational assistant. The team also included four adolescents with the role of monitors who participated as group leaders and liaisons between the group and the health team and two Lay Leds (young adults who are experts in the management of T1D, members of the Youth Center of Diabetes Association) who participated in the planning and development of the activities.

Comment 7: Authorization was obtained from the Ethics Committee of the Diabetic Association. Parents and participants signed informed consent.Was the focus group filmed? It states that “The interviews were transcribed in full, and significant nonverbal communication was recorded'’ Was this activity included in the consent form signed by the parents?

Response 7: Thank you for pointing this out. This activity was not included in the consent form, but, the parents consent verbally the team to take photos and make videos. At the beginning of the focus, we explained to the participants that we would make a video just for the study, and they agreed. 

Comment 8: The results are presented sequentially. Quotes from participants enrich the text. The numerical data in the figures are not clearly visible. The figures are not inserted in the previous text, they only have a “type” of title with the exception of figure 1. 

Response 8: We agree, changes were made and highlighted in yellow in the text. 

Reviewer 2 Report

Comments and Suggestions for Authors

Dear Authors,

Thank you for your interesting manuscript.  I find it important to report on experiences such as summer camps, but you should be very clear about what the scope is: To me, it sounds as if you are very into the subject, but I still have the impression that the aim of the paper is not clear enough. Why did you "implement" the Lay LeDU DM1 program?   will future camps run according to these experiences?

As it is a qualitative study you should state what your role in this camp was.  It doesn't seem that you as authors weren't present during the camp. Was it clear to participants that they would be interviewed at the end of the camp?

Some of you seem to have adopted the self-management education program from another program (spina bifida) with which you got a qualification. To me, it is a good thing but be aware that you should be very clear about which interest you have.  Could this be a bias? Please state openly how you dealt with this issue. (this can help to link with the paragraph in the discussion in line 398ff)

line 43: please introduce the readers to what the abbreviation LayLeDU DM1 stands for.

Line 96-97: in what way did you use COREQ?

Line 98 ff: it seems to be a convenience and not a purposive sample. Could you give some details on the summer camp and the association? Is this a regular activity of the association? Is this the only association in the region?
(members...)

Please describe also in more detail how the participants were selected. How many participants could have been participants but did not fulfil the criteria? Were you part of the selection team or the health team? Who (which professions) were part of the latter?

Does the association have an Ethics committee? How do you ensure independence? (line 107)

 In the results section: I suggest changing the colour of the figures to the "Positive aspects" which could be green and "negative aspects" red/or blue. This would be more intuitive.

Please reflect on your categorization of "positive" and "negative": Could at least the latter be named as "challenges"? Be aware of the formatting issues from line 167 onwards. (to line 243)

What did the two categories "emerge" (line 246) from? Isn't it more of a summary of themes/contents talked about in the interviews?

Could you please discuss the "tutoring system" you wrote about in line 389ff? Why do you think this was a negative aspect for participants in this study but not in the other one etc?

In the conclusions section: I suggest deleting the last paragraph as this may be your own purpose and not the one of your study. Maybe the suggestion I gave before to be clear on what your role and implementation purpose is, is to help as well here.

Thank you.

Author Response

The authors thank the editor and the reviewers for the suggestions and analyses that were accepted, hoping they can contribute to improving the quality of the manuscript.

Comment 1: To me, it sounds as if you are very into the subject, but I still have the impression that the aim of the paper is not clear enough. Why did you "implement" the Lay LeDU DM1 program?   will future camps run according to these experiences?

Response 1: This study enabled the development of knowledge regarding the design and implementation processes of Lay LeDU DM1. The focus groups were extremely relevant, as they contributed to understanding the value attributed to the program from the perspectives of the participants.

Comment 2: As it is a qualitative study you should state what your role in this camp was.  It doesn't seem that you as authors weren't present during the camp. Was it clear to participants that they would be interviewed at the end of the camp?

Response 2: The principal investigator participated in all activities in the summer camp. On the first day, she explained the objectives and the methodology for all the participants.  This information is on Line 136-137 “The principal investigator, a pediatric nurse specialist in child health, conducted the interviews, after taking part in all camp activities, having developed a relationship of trust with the adolescents”

Comment 3: Some of you seem to have adopted the self-management education program from another program (spina bifida) with which you got a qualification. To me, it is a good thing but be aware that you should be very clear about which interest you have.  Could this be a bias? Please state openly how you dealt with this issue. (this can help to link with the paragraph in the discussion in line 398ff)

Response 3: Compared to the study that inspired it, we consider it relevant to note that young people with Spina Bifida have a different chronic condition, being more dependent on self-care and experiencing locomotor limitations, which is not the case with adolescents with diabetes.

Comment 4: line 43: please introduce the readers to what the abbreviation LayLeDU DM1 stands for.

Response 4: We agree, changes were made and highlighted in yellow in the text. 

 Comment 5: Line 96-97: in what way did you use COREQ?

Response 5: The manuscript was written following the COREQ - Consolidated Criteria for Reporting Qualitative Research. We agree, changes were made and highlighted in yellow in the text. 

Comment 6: Line 98 ff: it seems to be a convenience and not a purposive sample. Could you give some details on the summer camp and the association? Is this a regular activity of the association? Is this the only association in the region?
(members...) Please describe also in more detail how the participants were selected. How many participants could have been participants but did not fulfil the criteria? Were you part of the selection team or the health team? Who (which professions) were part of the latter?

Response 6: For the participation in the camp it was a convenience sample, the team selected the participants who had difficulty managing their condition. However, for the focus group was a purposive sample, all of the participants who were in the camp, accepted participated in this study

Comment 7: Does the association have an Ethics committee? How do you ensure independence? (line 107)

Response 7:  It´s an independent committee, which includes professionals from various fields such as health, social sciences, humanities, law and also includes community representatives. Authorization was obtained from the Ethics Committee of Diabetic Association. Parents and participants signed informed consent.

Comment 8: In the results section: I suggest changing the colour of the figures to the "Positive aspects" which could be green and "negative aspects" red/or blue. This would be more intuitive.

Response 8: We agree, changes were made and highlighted in yellow in the text. 

Comment 9: Please reflect on your categorization of "positive" and "negative": Could at least the latter be named as "challenges"? Be aware of the formatting issues from line 167 onwards. (to line 243)

Response 9: We agree, changes were made and highlighted in yellow in the text. 

Comment 10: What did the two categories "emerge" (line 246) from? Isn't it more of a summary of themes/contents talked about in the interviews?

Response 10: We follow Bardin, so we used the terms that the author defined. If we used thematic analysis we could use themes or contents.

Comment 11: Could you please discuss the "tutoring system" you wrote about in line 389ff? Why do you think this was a negative aspect for participants in this study but not in the other one etc?

Response 11: They refer to the role of tutors were negative because tutors and tutees were in different dormitories, so it was difficult to plan together, they suggest in future camps, tutors and tutees could be in the same dormitories to facilitate the process.

Comment 12: In the conclusions section: I suggest deleting the last paragraph as this may be your own purpose and not the one of your study. Maybe the suggestion I gave before to be clear on what your role and implementation purpose is, is to help as well here.

Response 12: The focus groups contributed to understanding the value attributed to the program from the perspectives of adolescents and Lay Leds. In the beginning, we didn´t know what was the most important for them. The Lay Leds state that is important for them to be leaders and models of younger adolescents. The program motivates the adolescent bus and also motivates the Lay Leds to improve their management.

Reviewer 3 Report

Comments and Suggestions for Authors

The qualitative study would benefit from incorporating varied methods to enhance consistency and robustness, rather than relying solely on focus groups. Additionally, integrating a relevant behavioral theory to underpin your intervention program could provide a stronger theoretical foundation and improve the overall rigor and applicability of your study.

Author Response

The authors thank the editor and the reviewers for the suggestions and analyses that were accepted, hoping they can contribute to improving the quality of the manuscript.

Comment 1 _ Line 94 – Consider using alternative methods to enhance the precision of data collection

Response 1_ We agree, changes were made and highlighted in yellow in the text. 

Comment 2_ Line 95 – Some Behavioral theories should be used to support your methods

Response 2_ The educational intervention strategies employed in the program were grounded in Bandura's sociocognitive theory and emphasized key concepts such as self-efficacy, self-regulation, and observational/modeling learning.

Comment 3_ 107- The Ethics Approval number should also be presented

Response 3_ We agree, changes were made and highlighted in yellow in the text. 

Comment 4_110 – Please provide a description explaining why the groups were separated into male and female, including the rationale.

Response 4_ Male and female participants received the education program in separate groups. This strategy, derived from the original study, showed favorable results as it allowed adolescents of the same sex greater freedom to share personal experiences.

Comment 5_ 119- It seams that you used this study to evaluate the camp program. If this is the case, you should provide a detailed

Response 5_  The focus groups contributed to understanding the value attributed to the program from the perspectives of adolescents and Lay Leds.

Reviewer 4 Report

Comments and Suggestions for Authors

Dear authors,

thank you very much for this interesting work. The education and training of adolescents with T1D is an important topic, as many adolescents and young adults have difficulties taking on more responsibility in their own self-management while also "growing up", leading to suboptimal glucose control during this period and jeopardizing their longterm health. New approaches to mitigate these challenges are welcome. I think your manuscript would benefit from a more detailed description of the concept of transition, the program itself, of how you coded the interviews and a more concise conclusion, including your plans for the furture. This would also make your research more palpable for the readers. I have attached my specific comments and questions and I hope you find these helpful.

Best wishes

Author Response

The authors thank the editor and the reviewers for the suggestions and analyses that were accepted, hoping they can contribute to improving the quality of the manuscript.

Major Points

Comment 1_ Incorporate the concept of transition in the introduction.

Response 1_ We agree, changes were made and highlighted in yellow in the text

Adolescents experience a process of progressive transition towards autonomy as they develop and build autonomy. However, when the first episode of T1D occurs in adolescence, the adolescent is also experiencing a health/illness transition. At this stage of the study, we are considering the hypothesis that a summer camp with a self-management education program will help adolescents with T1D become autonomous in managing their disease, in accordance with Afaf Meleis' Transitions Theory.

Comment 2_ Introduction. Details of the program

The program is based on Kate Lorig tasks (Therapeutic management, Emotional management, and Role management) and Self-management skills: : Problem-solving - this is an essential self-management skill, which involves identifying problems and recognizing possible solutions, asking friends or health professionals for suggestions, implementing and evaluating the results; Decision-making - decision-making is part of the problem-solving process in the different stages of the illness and requires knowledge to support behavioral change and to make informed decisions; Finding and using resources - more than having the resources, you need to know how to use them; Training and establishing partnerships with the health team - with the change in the care paradigm, it is necessary to regard the health professional as a partner and not as a prescriber, and the team must also receive differentiated training to be able to promote self-management; Drawing up an action plan - the person must define achievable goals and objectives and operationalize the plans to achieve them.

Lorig developed their programs based on Social Cognitive Theory (Bandura). Line108-110 “The educational intervention strategies employed in the program were grounded in Bandura's sociocognitive theory and emphasized key concepts such as self-efficacy, self-regulation, and observational/modeling learning [16]”.

The program is conduced on small groups, male and female. Male and female participants received the education program in separate groups. This strategy, derived from the original study, showed favorable results as it allowed adolescents of the same sex greater freedom to share personal experiences.

The pairs are chosen by age, the older ones are tutors independent of their management skills.  Tutors and tutees are both motivated to improve their ability to help the tutees and also to be a model to other one.

Comment 3 _ details of analyzed data.

Response 3_ We agree, changes were made and highlighted in yellow in the text

Comment 4_Conclusion: next steps

Response 4_  We agree, changes were made and highlighted in yellow in the text

 Minor Points

Comment 1_ Why was it important?

Response 1_ We agree, changes were made and highlighted in yellow in the text

“In Portugal, the reviewed literature highlights the initiatives of the Portuguese Diabetes Association, which organized holiday camps for adolescents with T1D in recent years, with data from these camps disseminated in various reports [13,14]. However, there remains a significant gap in the available information regarding the impact of these camps on self-management.”

Comment 2_ Second Sentence

Response 2_ Thank You for pointing this. In this phase I was referring the results of Expert Patient Program, it was a citacion by Kate Lorig.

Comment 3_ Define Expert patients

Response 3_ We agree, changes were made and highlighted in yellow in the text. Line 51-53

Comment 4_ Malheiro

Response 4_ Yes, she is the same person. She developed a program for youth with Spina Bifida in Portugal and worked with me to implement the program in teens with T1D.

Comment 5_ What did you plan to do with this information?

Response 5_ The focus groups contributed to understanding the value attributed to the program from the perspectives of adolescents and Lay Leds.  With this information, we can develop knowledge to draw a new program.

Methods

Comment 1_ Why did you include Adolescents with HbA1c>8?

Response 1_ The recruitment was based on values during the year, if they have HbA1c>8 they need to improve their diabetes control. However, in the last evaluation, immediately before the camp, some of them had values <8, but they was already choised.

Comment 2_ Dividing by gender or sex.

Response 2__ We agree, changes were made and highlighted in yellow in the text.  we divided by sex (male and female)

Comment 4_ Grups with 16 adolescents

Response 4_ _ We agree, It was a limitation, but we had no time at last day of the camp to do groups with 8 adolescentes, because they must to do other activities planned by the diabetes association. 

Comment 5_ The interviews were conducted by nurses

Response 5_ The principal author, nurse and investigator were there and conducted the interview, the other nurse was there us an assistant, it was important because the adolescents had a good relashionship with him. 

 Results

Comment 1_ Age > or >15 and Disease 8

Response 1_ It was to organize the data, the average age was 15 and the duration of the disease was 8 years.

Comment 2_ You asked the participants about their opinion

Response 2_ We asked what they think about the program, and they talked about themes that they consider important and relevant about the program.

Comment 3_ Why did you choose to analyze participant and expert patients together

Because the expert patients valorized their role as expert patients, and the analysis of the program is very similar to the participants.

Comment 5_ Did the girls find it positive)?

Response 5_ The girls were a little bit shy and they preferred the division by sex to be more free to talk about their experiences.

Discussion

Comment 1 line 340-341

Response_ They liked but the activities are very programmed and they wanted to have some free time to choose some preferred activities.

Comment 2 line 389

Response 2_ Thank you for pointing this out. They refer to the role of tutors were negative because tutors and tutees were in different dormitories, so it was difficult to plan together, they suggest in future camps, tutors and tutees could be in the same dormitories to facilitate the process.

 Comment 3 line 403

Response 3_ We agree, changes were made and highlighted in yellow in the text (introduction)

“The expert patients underwent training before the camp and actively participated in the creation of case scenarios for role-playing exercises”.

Round 2

Reviewer 2 Report

Comments and Suggestions for Authors

Thank you for the revised manuscript, the "little" things adjusted or added contributed clarity.